# Innovation and Collaboration: Opportunities for the European Seaweed Sector in Global Value Chains

**Trond Selnes, Else Giesbers and Sander W. K. van den Burg *** 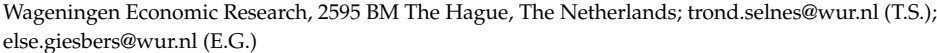

Wageningen Economic Research, 2595 BM The Hague, The Netherlands; trond.selnes@wur.nl (T.S.); else.giesbers@wur.nl (E.G.)

**\*** Correspondence: sander.vandenburg@wur.nl; Tel.: +31-0-70-3358129

**Abstract:** The European seaweed sector transitions from harvesting wild stocks only to harvesting and farming seaweed. This transition comes with the need to rethink the role of the European sector on the global scale; insight is needed into the organization of, and innovation in, the global seaweed value chain. This article presents results from our study on value chains using Gereffi's conceptual framework. A systematic review of scientific publications published between 2010 and 2020 was executed for five markets: pharmaceuticals, bioplastics, biostimulants, alginate and cosmetics. It is concluded that innovation in the use of seaweed takes place across the globe and thus that a focus on high-value applications alone will not set the nascent European seaweed sector apart from established producing regions such as Asia. The studied global value chains are organised around strong lead firms that require suppliers to produce according to codified product characteristics. The European seaweed sector needs to increase the collaboration and develop joint efforts to develop safe and sustainable products that meet the demands of regulators, lead firms and consumers. Stronger coordination in the value chain will facilitate further business development, by stimulating collaboration and innovations.

**Keywords:** seaweed; value chain; collaboration; innovation; governance

## 1. Introduction

Seaweeds are increasingly becoming an interesting biomass for a range of sectors, such as pharmaceutics, bioplastics, biostimulants, alginates and cosmetics. Most of the production and processing is currently concentrated in Asian countries [1] but the European sector is working hard to enhance its role in the seaweed value-chain. Macroalgae production is still relying on the harvesting of wild stocks (68% of the macroalgae producing units) but macroalgae aquaculture (land-based and at sea) is developing in several countries in Europe and is currently representing 32% of the macroalgae production units [2]. France, Ireland, and Spain are the top three countries in the number of macroalgae production units. Algae production in Europe remains limited by a series of technological, regulatory and market-related barriers [3].

In the increasingly globalized economy, it is inevitable that the nascent European seaweed sector will face competition from established producers [4,5]. Following Tallman et al. [6], the competitiveness in global value chains is not a matter of having the lowest production costs, but a matter of generating sufficient value. To this end, upgrading strategies are essential, i.e., to move to higher value activities and market segments [7].

Competitiveness in global value chains can be reached through a 'low road' (fight competitors based on low costs) and a 'high road' which, in this last case, comprises innovation, increasing productivity, and enhancing quality [8]. The high road requires the use of upgrading strategies to increase the value of produced materials. This upgrading strategy is explicitly sought for in the GENIALG project (https://genialgproject.eu/ Last accessed on 23 June 2021), which aims to develop seaweed-based products for a variety of markets, with a focus on *Saccharina latis*sima (Linnaeus) and *Ulva* spp. *Saccharina latis*sima

is a brown seaweed known as sugar kelp or kombu royal in Europe. *Ulva* spp., known as sea lettuce, is a group of green seaweeds, also presenting high biomass production yield and showing high farming expansion potential in Europe [9].

The objective of this article is to evaluate the prospects for European seaweed sector to gain a foothold in the global seaweed value chain, with a focus on the applications developed in the GENIALG project. The main research question guiding this manuscript is: "what does the current organisation of the global seaweed value chains mean for the European seaweed industry?". This main research question is broken down into the following sub-questions:

- What concepts can be used to describe the organisation of global value chains?
- How can the different value chains for seaweed be characterized?
- What are the consequences for the future of the European seaweed industry?

## 2. Materials and Methods

### 2.1. Conceptual Framework

The Global Value Chains framework is used to study how global industries are organized by examining the structure and dynamics of different actors involved in a given industry [7]. A value chain is defined as the "full range of activities that firms and workers do to bring a product from its conception to its end use and beyond" (ibid., p. 7) and typically includes a variety of activities: design, production, marketing, distribution and support to the final consumer. These activities can be performed within the same firm or divided among different firms—emphasized by the notion of a chain. The fact that they are increasingly spread over different countries and continents explains why the value chain is regarded as "global" [10].

A focus on the value chain allows us to describe the full range of activities of a product from its conception to end use and beyond, as argued by Gereffi and Fernandez-Stark [4]. In this way, we can study the structure and organisation of the global industry in question and trends that shape change. Following the approach developed by Gereffi and Fernandez-Stark (ibid.) and Gereffi et al. [11], the global seaweed value chain is here described along 3 dimensions: innovation, geographic coverage and value chain governance. Table 1 below indicates how each of these dimensions is operationalised in this study.

**Table 1.** Operationalisation of the three dimensions (Based on Gereffi & Fernandez-Stark [7]).

| Dimensions | Operationalisation |
| --- | --- |
| Innovation | Quantitative analysis of trends in number of scientific literature |
| Geographic scope | Identifying the spread in location of publications for the five value chains |
| Governance | Trends in (the authority) relations, in relation to Gereffi's fivefold distinction (see Figure 1 below) |

We follow Gereffi [12] (p. 97) and his definition of governance as "authority and power relationships that determine how financial, material and human resources are allocated and flow within a chain". The Global Value Chains literature identified five types of governance structures (see Figure 1) each with a different combination of (1) the complexity of the information shared between the actors for the transactions; (2) the ability to codify transactions; (3) the competences or capabilities of the suppliers [7,10,13].

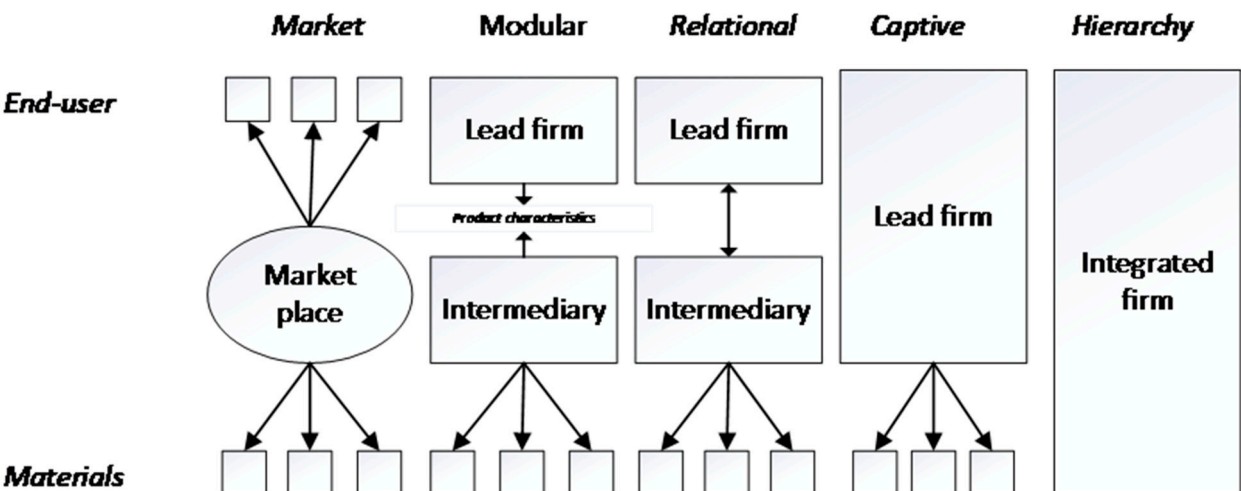

**Figure 1.** Five types of value chain governance (Based on Gereffi et al. [11]).

*2.2. Operationalisation*

Following the objectives and structure of the GENIALG project, the analysis presented below focused on five value chains where there is potential for an increased use of seaweed. Given the fact that the level of technological development differs per value chain, and that the potential of seaweed is not equally well understood, further scoping is done where possible.

For the pharmaceuticals value chain, the interest goes out to the extraction of fucoxanthin for medicinal use, mainly anti-cancer. The regular medical circuits are addressed, omitting the use of seaweed in the alternative medical circuits.

With respect to bioplastic, the analysis focused as much as possible on the production of acrylic acid from seaweed, given the interest of the GENIALG consortium partner. However, due to the lack of information specific on acrylic acid, our analysis at times focused on bioplastics in general.

For biostimulants, the focus is on the use of biostimulants for plants and animals. The analyses of alginate and ulvans value chains, and cosmetics value chains do not focus on a specific application of alginate or ulvans or a specific cosmetic application.

There are multiple search engines, with the Web of Science and Scopus being the two world-leading citation databases [14,15]. We chose Scopus due to its reputation as being more interdisciplinary, while the Web of Science is being stricter in peer-reviewed results. Pranckuté [14] concluded that Scopus is better suited due to a wider and more inclusive content coverage, the availability of profiles for all authors, institutions, and serial sources, being one single database without the confusion or additional restrictions regarding content accessibility and it is more open to the society (free access to author and source information, including metrics). The choice of the specific compounds fucoxanthin, alginate or acrylic acid are due to a choice made by the project consortium in an early stage of the project. In evaluating the subject areas in which scientific papers are published, we used the standard categorization of Scopus (See this link for the complete overview of subject areas Scopus uses to categorise articles: https://service.elsevier.com/app/answers/detail/a_id/15181/c/10547/supporthub/scopus/ Last accessed on 28 May 2021).

*2.3. Data Collection*

A systematic literature search was performed on 28 May 2021. Included are articles published between 1990 and 2020. The year 2021 was excluded because it would give an incorrect overview since the year 2021 was not completed at the time of analysis. To be as complete as possible and be able to give a comprehensive overview, we used different search queries for each of the five value chains. For every value chain, a search was performed using the search terms 'seaweed', 'macroalgae', 'macro-algae', 'brown

macroalgae', 'brown macro-algae', 'green macroalgae', 'green macro-algae', 'Saccharina latissima' and 'Ulva'. For some value-chains, multiple searches were performed with additional search terms to get a more specific overview on the specific applications, for example on 'pharmaceuticals' and 'fucoxanthin'. An overview of the number of articles we found using the specific search terms can be found in Appendix A. Appendix B shows all search queries that are used, corresponding to the figures in the text.

For each combination, we specifically looked into the number of articles published each year, what countries the authors came from and in what subject field the articles were published according to Scopus. All papers retrieved were included in the analysis, including review papers.

## 3. Results

### 3.1. Pharmaceuticals

The pharmaceutical industry is a trillion-dollar industry that is still growing; cancer research is one of the important targets for the research. The interest in the effects of fucoxanthin on cancer cells is related to its expected anti-proliferative and distinctive and possibly unique effect. Fucoxanthin from seaweed is currently seen as a major opportunity for future pharmaceutical applications as fucoxanthin is considered to be a powerful antioxidant with many attractive features, such as being antibacterial, anti-obesity, anti-diabetes, anti-inflammatory, antidementia, possibly important for brain injury recovery and a great potential for being an anticancer agent [16–20]. Wang et al. [21] also argue that combinations with other drugs could enhance the effect of both fucoxanthin and the other drug or reduce the dose without reducing the effect, which may create more effective and less harmful therapeutic strategies. Besides, fucoxanthin showed no toxicity [21]. At the same time, Lu et al. [22] also point to great challenges related to growth rates, the fucoxanthin content, product quality and potential environmental effects which might compromise the development of cost-effective production methods [16]. Much work is still to be done and is being done due to the high potential of these marine compounds [20,21].

#### 3.1.1. Innovation

When we look at the number of publications on seaweeds and pharmaceuticals in general, we see a major increase from 2007 and onwards. In the period 1990~2007 the number of publications never exceeded 5, whereas after 2007 the publication numbers raised to 114 in 2020, as can be seen in Figure 2. Publications on seaweed and fucoxanthin in pharmaceuticals remained low.

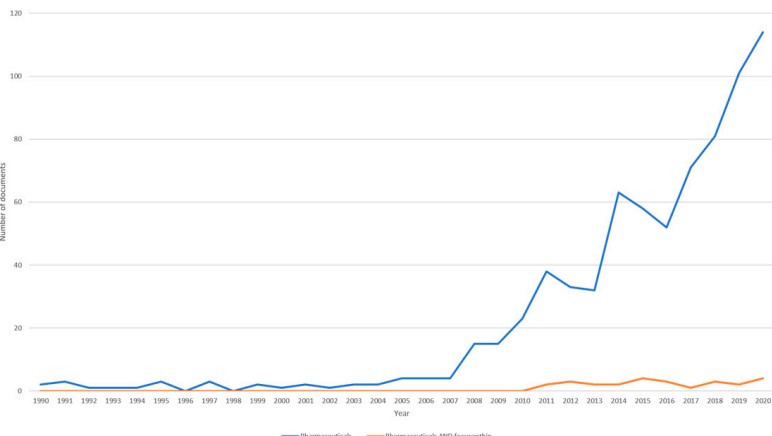

**Figure 2.** Documents by year for "pharmaceuticals" and "pharmaceuticals and Fucoxanthin" (Appendix B—i) (The superscript numbers listed below the figures refer to the search terms as mentioned in Appendix B).

The focus on fucoxanthin is a recent one but despite a limited number of studies, the interest is rising slightly. In science, the number of publications started to increase the last decade. From 1990 through 2010 there were not any publications. Since then, the number of publications has varied from 2 to 4 per year, with only one in 2017.

A look at the subject areas of the publications on pharmaceuticals and Fucoxanthin reveals that most attention is devoted to health issues as pharmacology, toxicology and pharmaceutics (20.0%) and biochemistry, genetics and molecular biology (17.8%). Agricultural and biological science also hold 20.0% of the share in published articles.

### 3.1.2. Geographic Coverage

When it comes to the production of seaweed, Asia dominates with 99% of the total seaweed production, captured and aquaculture [23]. If we look closer at who is studying the effects of Fucoxanthin on cancer cells in Figure 3, we see again that Asia dominates. In particular, we see that South Korea is in the lead when it comes to scientific publications, with Iran at some distance. Then, Taiwan, Spain, Poland, Japan and India are following. These countries are the only ones with more than one publication and therefore the only countries that are listed in this graph.

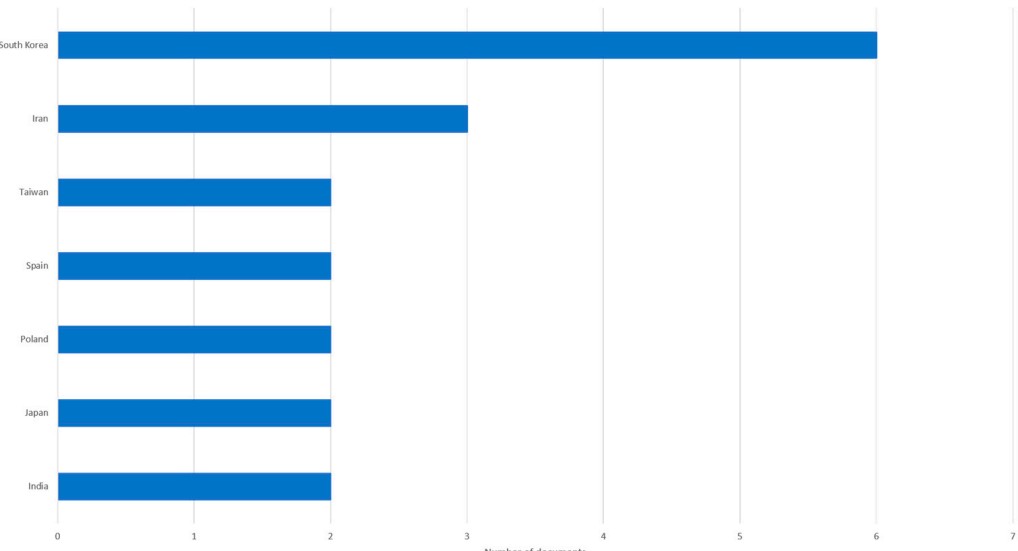

**Figure 3.** Documents by country or territory for pharmaceuticals and fucoxanthin (Appendix B—ii).

### 3.1.3. Governance

The pharmaceutical value chain was originally dominated by hierarchy with vertically integrated technology-oriented companies operating from their own country involving little international investments [24]. In 2006 scholars observed that much had changed [25] with (i) more globalisation instead of a country by country approach; (ii) Alignment of regulations in the EU, USA and Japan for better use of R&D; (iii) Increasing outsourcing of data management to India and China; (iv) Companies purchase each other and merge at a rapid rate (rapid spread of best practises but also power struggles and defending one's own turf); (v) Ever more complicated and complex analysis becomes the standard (comparing to an existing drug instead of comparing to a placebo); (vi) Molecular studies are blossoming at an exceptional speed as a springboard to new products.

Pharmaceuticals is now an even more complex value chain where the specific steps and requirements differ between types of medicine, manufacturers and countries [26]. The pharmaceutical industry uses 'tight' forms of coordination like the captive models [27].

We then will have a different value chain for the seaweed harvesting, processing and extracting process and the pharmaceutical production. The pharmaceutical industry will however keep its tight control of the quality. Besides, in the EU there are strict rules for testing and producing for instance cancer medicine, and the companies must meet the

regulatory demands of the EU CTR, the Clinical Trial Regulation. The commercial step also comes with demands concerning safety lifecycle management quality.

*3.2. Bioplastics*

A rapid growth of the super absorbent polymers market and an increasing demand for acrylic adhesives in high-end applications that require high shear and tensile strength along with shock absorbent characteristics is expected to propel the demand for acrylic acid in the years to come (https://www.transparencymarketresearch.com/acrylic-acid-market.html Last accessed on 23 June 2021). Seaweed is a candidate for use in the production of acrylic acid and by that fulfil the increasing need from the market. However, fossil fuel and plastic production are currently integrated [28], as most acrylic acid still derives from non-renewable petrochemical products. However, carbon rich plastics also come with high $CO_2$ emissions and about 80% of manufactured plastic accumulates as waste in landfills and natural environments, presenting an increasing hazard [28]. At the same time, the highly volatile prices of raw (petrochemical) material and increasingly stringent environmental regulations are expected to restrain growth of the (petrochemical based) acrylic acid market. As a result, the attention shifts to biobased renewable acrylic acids where biodegradable and bio-based plastics present a viable and attractive alternative [28].

3.2.1. Innovation

As Figure 4 indicates, prior to 2006 no year saw more than 2 publications for bioplastics or acrylic acid. Additionally, after 2006 the numbers varied from 1–4 publications. The year 2019 and 2020 were an exception with 8 and 7 publications, respectively, it remains to be seen whether this is a trend. In comparison, a search on just acrylic acid gives 59.437 hits, of which 2.354 in the year 2020. Seaweed-based acrylic acid is therefore still a rather marginal activity in the world of acrylic acid, at least measured in the number of publications.

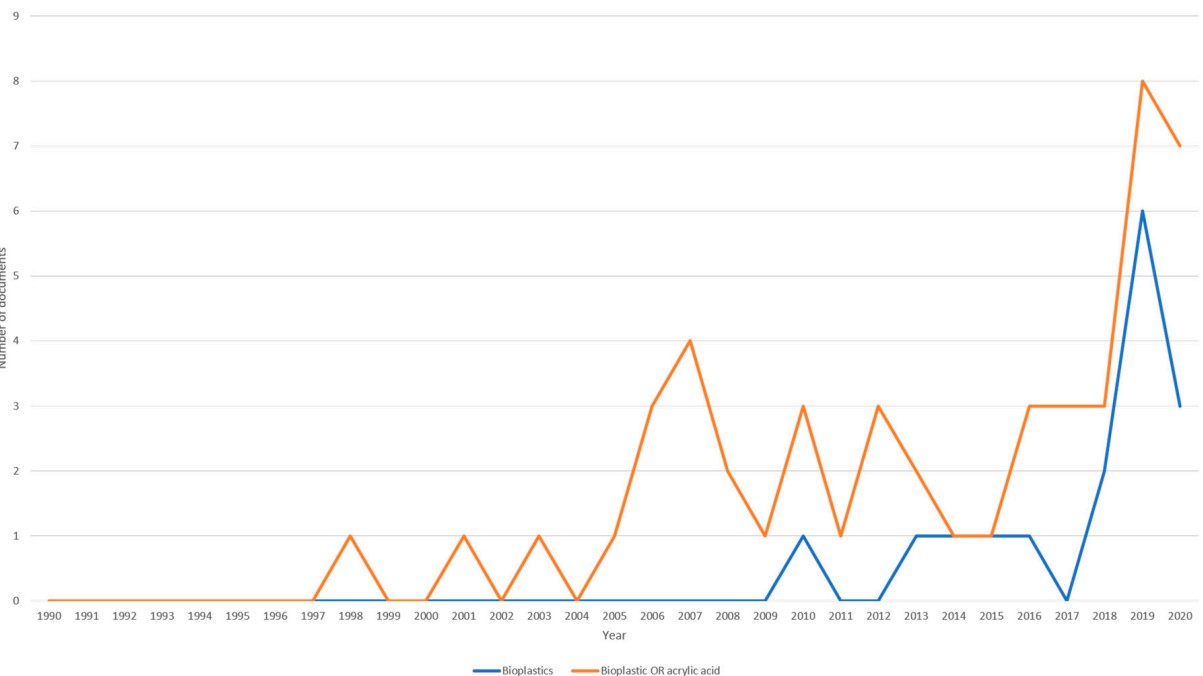

**Figure 4.** Documents per year for 'bioplastics or bio-plastics or acrylic acid' and 'bioplastics or bio-plastics' (Appendix B—iii).

Efforts to innovate in production and products of algae-based bioplastic using acrylic acid are now being conducted in various subject areas. Agricultural and biological sciences form the largest part, then comes environmental sciences. Chemistry and other areas comprise less than 10% of the publications.

### 3.2.2. Geographic Scope

Asia Pacific is expected to remain the leading market for acrylic acid in terms of demand with an estimated 47% share of the total volume consumption in 2018. Asia Pacific is also expected to be the fastest growing region over the next six years, due to the rapidly growing construction industry in emerging economies such as China, India and South Korea (https://www.transparencymarketresearch.com/acrylic-acid-market.html Last accessed on 23 June 2021). North America was the second largest market for acrylic acid followed by Europe, in 2012.

Considering the geographic spread of publications, we see that the United States, United Kingdom and China are ahead of the others, with the United States clearly in front. From Asia also Indonesia, Japan and Malaysia and India are active. Of the European countries we see Italy and Germany in addition to the United Kingdom. Only the countries with at least two publications are shown in Figure 5.

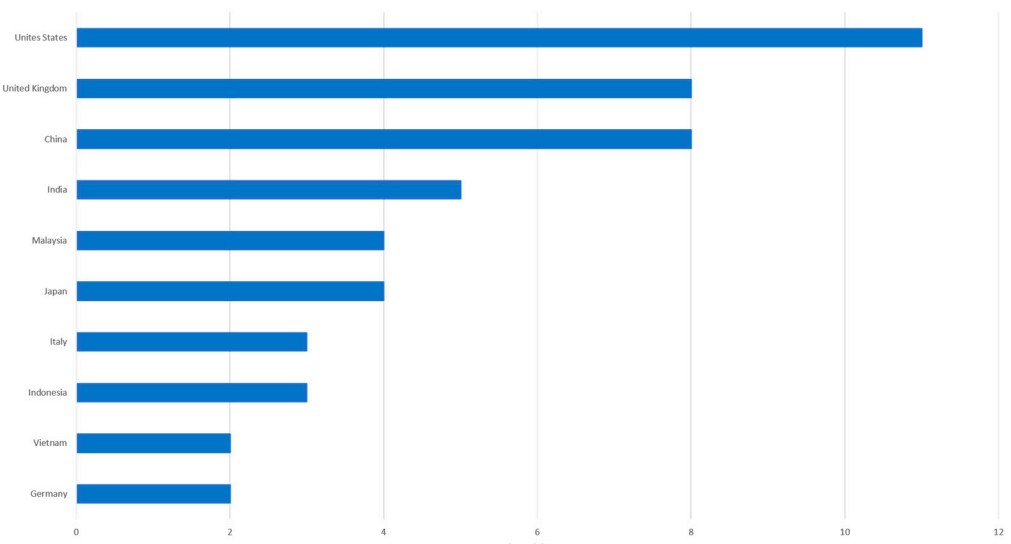

**Figure 5.** Documents by country or territory for bioplastics or bio-plastic or acrylic acid (Appendix B—iv).

### 3.2.3. Governance

The acrylic acid value chain ranges from the sourcing of raw materials (propene) to the delivery of the final product to end-use industries such as surface coatings, adhesives & sealants, plastic additives, surfactants, and also others (https://www.marketsandmarkets.com/Market-Reports/acrylic-acid-market-683.html#:~:text=Ltd.,key%20manufacturers%20of%20acrylic%20acid Last accessed on 23 June 2021). The Acrylic Acid Market report says that the industry is dominated by a few big players and fully integrated companies, with most of the key suppliers being forward integrated and market their products mostly through local and regional distribution channels. In addition, the report states that production units in North America and Europe supply acrylic acid globally, in particular to the Asia-Pacific region. In every country however, a few local players contribute substantially to the regional market as they can have bigger market shares in their region compared to the global players.

### 3.3. Biostimulants

The biostimulant value chain consists of a wide variety of products, derived from various resources. Agricultural biostimulants include diverse formulations of compounds, substances and micro-organisms that are applied to plants or soils to improve crop vigour, yields, quality and tolerance of abiotic stresses (http://www.biostimulants.eu Last accessed on 23 June 2021). Du Jardin [29] defines plants biostimulants as "any substance or

microorganism applied to plants with the aim to enhance nutrition efficiency, abiotic stress tolerance and/or crop quality traits". Various categories of biostimulants are subsequently defined by Du Jardin [29] (p. 4), of which seaweed extracts and botanicals is one.

The market for biostimulants is diverse and fragmented. Reliable data on the volume of the markets—and trends therein—is scarce [30]. The total size of the European biostimulant market is estimated at EUR 560 million in 2016, of which EUR 194 million is for seaweed extracts [31].

### 3.3.1. Innovation

In ancient times, seaweeds were already used seaweeds as sources of organic matter and fertilizer. The use as biostimulant is of more recent origin, yet there is somewhat of a history. Khan et al. [32] already concluded that seaweeds and products derived from seaweed were widely used because of plant growth stimulating components, but also that there is a lack of scientific data. At the time of writing Khan et al. [32] could list 25 different commercial seaweed products used as bio-stimulant.

The analysis of the number of scientific publications over time, see Figure 6, shows a rise in interest over the last decade, and in particular from 2014.

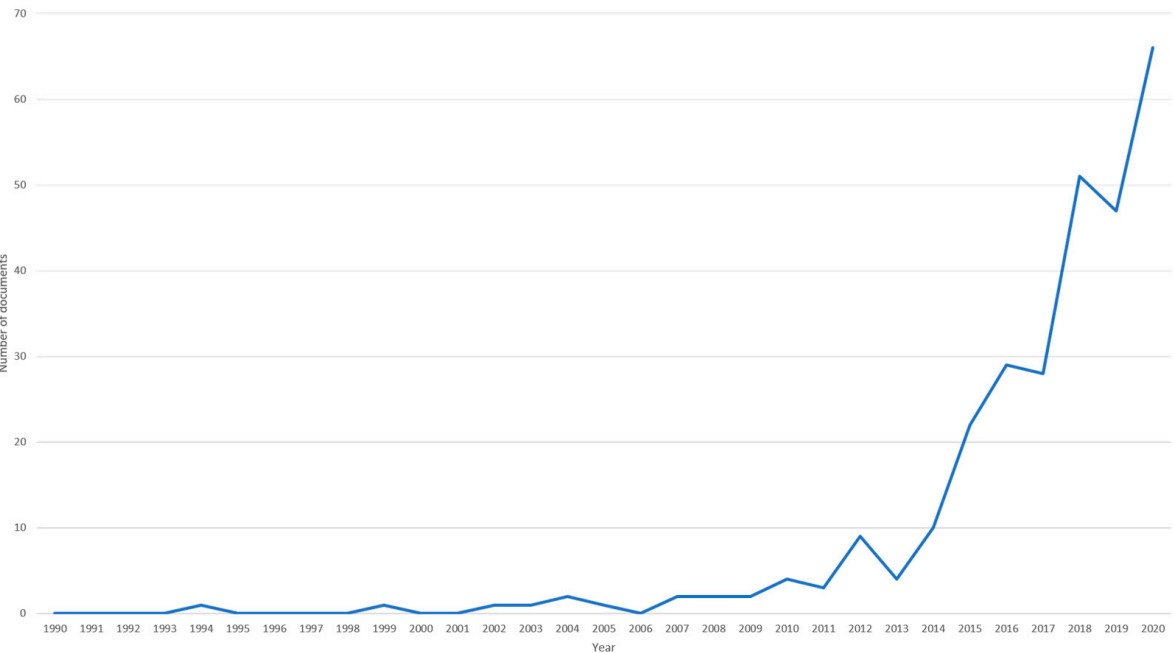

**Figure 6.** Documents per year for biostimulant (Appendix B—v).

Most of these publications (56.2%) are part of the subject area "Agriculture and Biological Sciences", confirming the interest in the use of biostimulants for improving agricultural production.

Recent investigations into the use of seaweed as biostimulant seek to understand the modes of actions by which seaweed extract acts as biostimulant (see, e.g., [33]) and explore new applications. For example, the potential to use seaweed extracts for enhanced cultivation of seaweeds is described by Hurtado and Critcheley [34]. In cases of crop damage, diseases but also changing physical conditions, the use of seaweed biostimulants can enhance seaweed production. Umanzor et al. [35] conclude that the use of seaweed extract can enhance thermal tolerance and growth of cultivated seaweeds.

### 3.3.2. Geographic Scope

Many of the players are located in France, Norway and Ireland. Additionally, countries with a strong agricultural sector such as Italy and Netherlands are home to some of the

companies. The French biostimulant value chain is expected to reach € 150 million in 2020, with seaweed extracts corresponding to approximately 1/3 of the total market [30]. This exceeds the projected UK market, estimated at EUR 100 million in 2022 [30]. Projections on other countries are not available.

Research and innovation in biostimulants take place across the globe, the Top 3 of countries with most publications being in Poland, Indonesia and Italy (see Figure 7).

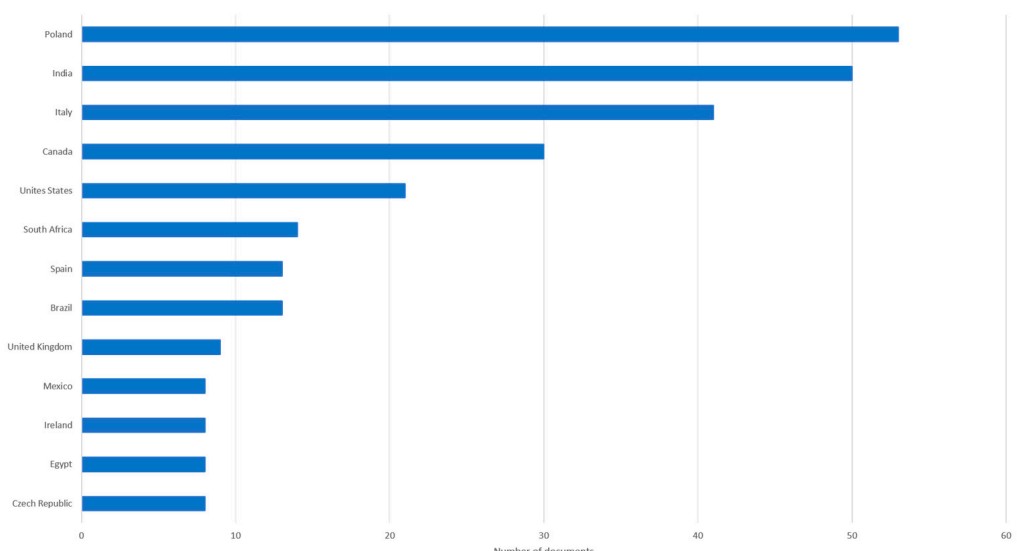

**Figure 7.** Documents by country or territory for biostimulant (Appendix B—vi).

Only the countries with a minimum of 8 publications are shown in this graph.

### 3.3.3. Governance

In the biostimulant value chain, the degree of explicit coordination and power asymmetry is low. The value chain is characterized by the presence of multiple companies in all steps in the value chain, from harvesting to sales. These companies generally all work in one step in the value-chain; there are no large integrated companies. Following the characterization as developed by Gereffi et al. [11], the global biostimulant value chain resembles the "market" type.

### 3.4. Alginate

The extraction of alginate from seaweeds is established practice, catering to various markets. Alginate is a commonly approved additive as emulsifying and thickening agent; E-numbers E400 to E405 refer to different alginates (http://www.food-info.net/uk/e/e400-500.htm Last accessed on 23 June 2021). Bixler and Porse [36] estimate that in 2009, approximately half of total alginate production was high grade, used in the food/pharmaceutical market, and the other half industrial grade for other applications. They also argue that there is a shift in species-in-demand, from *Ascophyllum nodosum* and *Macrocystis pyrifera* towards kelp species. Consequentially, as some these species prefer different geographical regions, global trade patterns of seaweeds for alginate production have changed.

### 3.4.1. Innovation

The number of scientific publications on alginate and ulvan shows growth since 2009 as Figure 8 illustrates. Before 2009 the annual number of publications was about 10, in 2020, 137 scientific publications on alginate or ulvan were published.

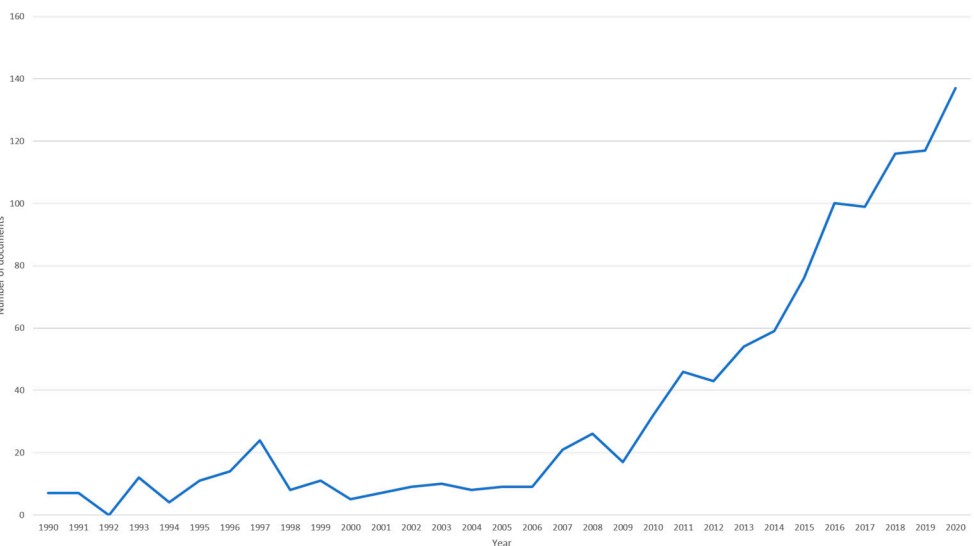

**Figure 8.** Documents per year for alginate or ulvan (Appendix B—vii).

The analysis of the subject area to which these publications belong confirms that alginate has a broad range of applications. Whereas agriculture and biosciences is the largest, it only represents 16.7% of the total publications. Other large groups include biochemistry, chemistry and chemical engineering.

The scientific literature shows a continuous search for new applications of alginate, all based on its capacity to encapsulate and transport other substances. Applications of alginate include pharmaceutical use, use in cosmetics and textile printing. In the pharmaceutical industry, alginate is used to improve drug delivery through controlled-release systems [37] and also for tissue recovery [38]. In cosmetics, alginate can be used as a carrier for fragrances [39] or oils [40].

### 3.4.2. Geographical Scope

It is well known that the production of seaweeds and alginate is concentrated in Asia. A closer look into the number of publications per country shows that of the top five countries (most publications published), four are Asian countries. Figure 9 shows that France completes the top five.

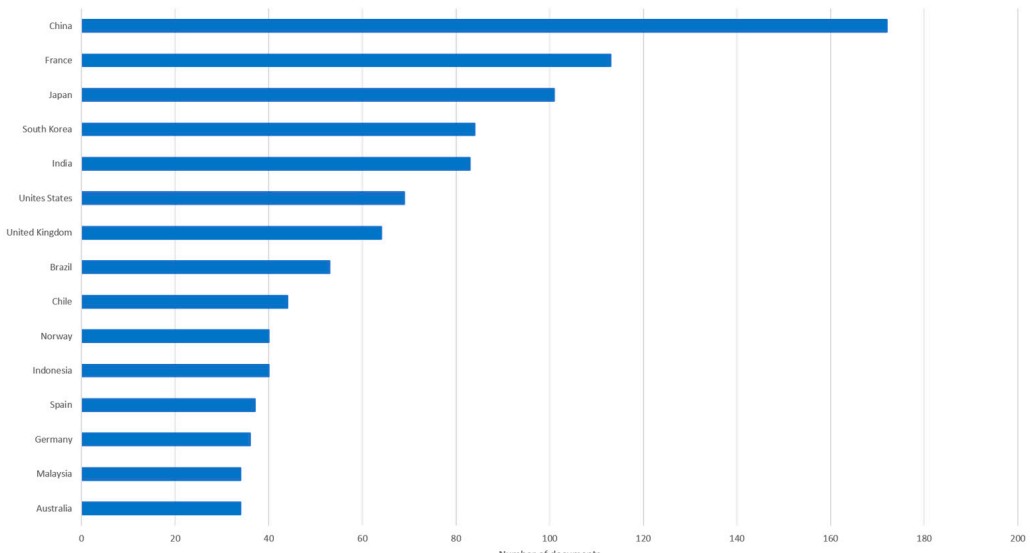

**Figure 9.** Documents by country or territory for alginate or ulvan (Appendix B—viii).

Only the countries with at least 34 publications are listed in this graph.

The concentration of alginate manufacturing in China is an ongoing process and nearly all industrial grade material was produced there, whereas Europe is the leading food and pharmaceutical-grade alginate production centre, but to some degree, China has already caught up [41].

### 3.4.3. Governance

As Bixler and Porse [36] argue, the alginate industries have benefited from the consolidation of alginate producers. The availability of seaweeds has not been a problem for alginates during the early 2000s but can be expected to present supply and cost problems, particularly for high gel strength types, in the future.

Looking at the organisation of global value chain for alginate—produced from brown seaweeds—it is clear that the global market for brown seaweeds is dominated by a few large companies that cover various steps in the supply chain. These companies produce alginate, for further application by other companies that produce final consumer products. The value chain type resembles the modular value chain type.

In contrast to the tropical seaweed value chain—where a large base of small-scale producers still exists—the production (harvesting and cultivation), processing and sale of brown seaweeds for alginate is dominated by a limited number of large companies. These exert influence over all steps of the value chain.

### 3.5. Cosmetics

The cosmetic industry is an industry with a globalised character and a worldwide worth of more than EUR 425 billion [42]. Since more consumer awareness is given to health and environmental issues [43], the industry wants to meet their goals to reduce their environmental footprint. Besides environmental reasons, the sector is also looking for new, natural ingredients to reduce the use of materials like plastic microbeads for marketing reasons (https://cosmeticseurope.eu/how-we-take-action/driving-sustainable-development/ Last accessed on 28 May 2021) [42,43]. Seaweeds are an interesting alternative because of their non-irritating, non-toxic, anti-inflammatory, antioxidant and UV-protection characteristics [42,44]. Extractions from seaweeds, especially from brown and red seaweeds, are therefore used in many different cosmetical products like sunscreen, moisturizers, anti-aging products, whitening and hair care [45]. Because of the multiplicity of products, lots of different types of seaweeds can be used.

### 3.5.1. Innovation

Innovation is a key aspect of the lucrative and fast-paced cosmetics industry, as result of consumers' ever-changing demand and the short life cycles for cosmetics products. Innovation is seen as the key to success for businesses [46]. In Europe, there are at least 77 scientific innovation facilities carrying out research in this field. Large industry players tend to have multiple research centre that focus on product development, market research or regulatory compliance [47] (p. 27).

Innovations in this industry are strongly driven by the desires of the consumer to have new, better and safer cosmetical products wherein environmental and ethical issues become more and more important [42,48]. Over the last years, many investments focused on developing alternatives of testing on animals. Right now, reducing the environmental footprint through environmental efficient techniques, decreasing waste and decreasing emissions throughout the sector are one of the main striving points (https://cosmeticseurope.eu/how-we-take-action/driving-sustainable-development/ Last accessed on 28 May 2021).

Up until 2009, not more than 5 articles about cosmetics and seaweed have been published in one year. After 2009, there is an upward trend in the number of publications, as can be seen in Figure 10 below. In 2020, a total of 80 articles have been published about

this topic. Most of the articles focus on agricultural and biological topics and characteristics of seaweed.

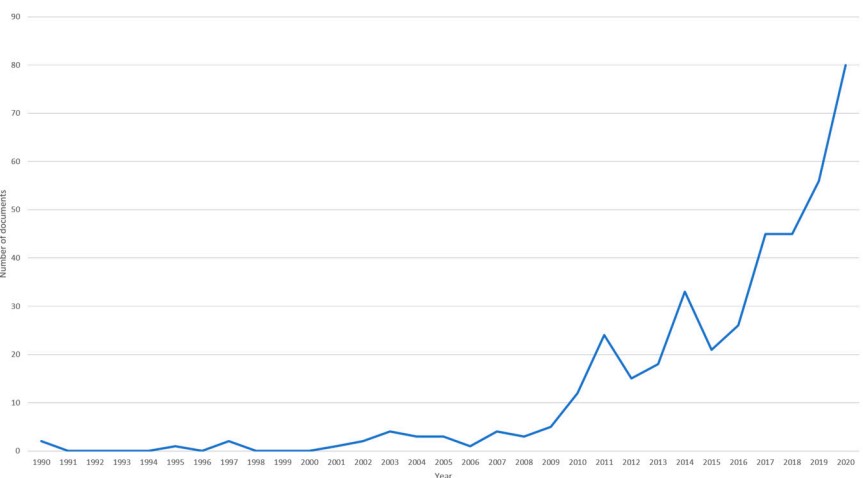

**Figure 10.** Documents by year for cosmetics (Appendix B—ix).

### 3.5.2. Geographic Scope

The three countries that publish the most articles are Asian countries (India, South Korea and Indonesia). Other countries that have published multiple articles are spread all over the world as can be seen in Figure 11 below. The figure only shows countries that have published at least 11 articles.

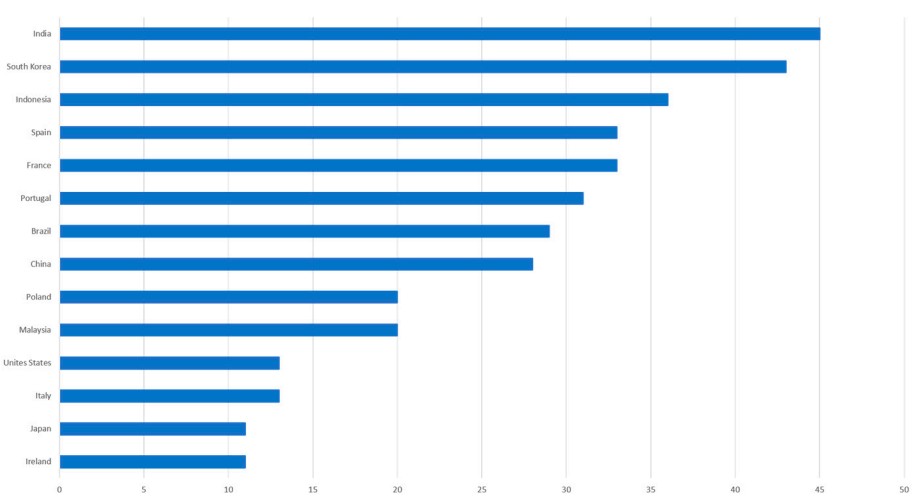

**Figure 11.** Documents by country or territory for cosmetics (Appendix B—x).

Currently, most large cosmetic brands are located in Europe, the United States and Japan (https://brandfinance.com/wp-content/uploads/1/cosmetics_50_free.pdf Last accessed on 28 May 2021).

### 3.5.3. Governance

For most cosmetic companies, the use of seaweed is a side-issue. Some examples can be found in which relatively small companies both cultivate algae and produce cosmetical products. However, most cosmetic companies do not produce seaweed themselves but buy seaweeds that are cultivated and harvested in Asian waters [47]. The cosmetic value chain is, especially in Europe, fragmented; existing of many small and medium-sized enterprises (SMEs). The market in the United States and Japan is however dominated by a few large companies. Large conglomerates cover over half of the worldwide cosmetics market [49].

Most of the companies are active in one part of the value chain of seaweeds for the cosmetical industry. Regulation of the value chain mainly focuses on the use of certain ingredients in cosmetical products and information given to consumers. This value chain can be categorised as a modular type.

## 4. Discussion

### 4.1. Comparing the Five Value-Chains

In all of the five value chains studied, an increase in the number of scientific publications started in the early 2010s. In absolute terms, the number of scientific publications is greatest in the value chains pharmaceuticals and alginate.

An analysis of the geographical spread of these publications does not point to one or two dominating countries. Instead, a group of countries are responsible for most publications, including (in alphabetical order) Brazil, Canada, China, France, India, Indonesia, Italy, Japan, Poland, South Korea, Spain, Taiwan and the United States. For seaweed in cosmetics and alginate Asian countries dominate, just as for the pharmaceutical use of fucoxanthin in cancer research. For biostimulants the picture is much more mixed.

Large companies operate in the 4 out of 5 value chains studied, the exception being the biostimulant sector. Such large companies have considerable influence over the other segments of the value chain and can impose demands, standards and private regulation. Additionally, we see that public regulation plays an important role in these markets, and that new regulation, in particular on environmental quality standards, is in the making. The pharmaceutical industry is the most densely regulated and strict sector. Although regulation is less strict and less organised for the other value-chains, complex regulatory networks are in place that influence the development and marketization of seaweed-based products.

Looking at governance in the value chains, it is noteworthy that, following the typology of Gereffi et al. [11], various types kind of governance occur, but that a hierarchy with large firms dominating both the seaweed and the higher parts of the value chains is absent. With an exception for the biostimulant sector, the value chains are characterized by strong lead firms that require suppliers to produce according to codified characteristics. The alginate value chain for instance is of a modular type, with a central role allocated to the alginate producers—which delivers to companies further along the value-chain. The pharmaceutical and the bioplastics value chains are captive types and in particular the pharmaceutical value chain exercises tight control.

### 4.2. Implications for European Seaweed Sector

The European seaweed sector is a young one with the eyes towards the future. Although the opportunities of bulk production in Europe should not be ignored, it seems likely that most businesses will be oriented towards a quest for entering the higher segments in the market; the high value products for the pharmaceuticals, bioplastics, biostimulants, alginate and cosmetics value chain. A market strategy focusing on high value products might imply that Europe could import more raw material from low-cost countries and then add value in Europe. The question is then whether the origin and quality standards of the import can be guaranteed. Stricter regulations and quality demands from the market value chain could reduce the likelihood of importing much seaweed from low-cost countries as the European businesses must be sure of the quality of the raw material. The expectation in Europe is that more regulation on safety and environment will be implemented, which will call for more attention to demonstrating effects/impacts on the entire value chain. For example, the pharmaceutical industry (see Section 3.1.3) prefer tight control and high quality standards. European companies might be able to ensure the origin of the extracts delivered but will the European businesses also excel in quality and innovation? Innovation is key to find applications with fucoxanthin, bioplastics or cosmetics, but innovation is not confined to Europe as it takes place in many countries and continents. Investments in unique selling points could ensure a differentiation of products and the deliverance of unique, sustainable, certified products. This could set European producers apart.

Over the last years, European governments, companies and research institutes have invested in making seaweed cultivation work and in research into new applications. There are promising options but the next challenge is to develop marketable products, bringing benefits to the European seaweed businesses and by that stimulate more demand (market pull). Our study points to a need for coordinated investments in the seaweed value chain. Such a strategy should help facilitate further business development by stimulating collaboration and innovations, linking various segments of the future seaweed value-chains. Joint efforts to further develop seaweed value-chains enables a shared effort to develop clear regulatory frameworks, addressing topics such as human and environmental safety standards, developed and supported by public and private actors.

## 5. Conclusions

This paper uses the framework developed by Gereffi and others [7,11,12] to characterize the global seaweed value chains. Key concepts in the analysis include innovation, geographical coverage and governance. These are used to illustrate the differences in the organisation of five seaweed value-chains: pharmaceuticals, bioplastics, biostimulants, alginate and cosmetics. Results are summarized in Table 2.

**Table 2.** Summary of the value chain analysis.

| | Innovation | Geographic Scope | Governance | |
| --- | --- | --- | --- | --- |
| | | | Description | Type |
| Pharmaceutical | Growth number of publicationsFocus on fucoxanthin is recent | Many Western companies but for research mixed (more Asian) | Many large companies; tight control, strict regulation EU. | Captive |
| Bioplastics | Few publications | Asia dominates | A few big companies. No clear role for regulators, but policies are underway. | Captive |
| Biostimulant | Growth number of publications. Most cited publications on plant biostimulants | Globally distributed | Mix of large and small companies. Some government regulation. | Market |
| Alginate | Growth no. of publications. Mixed topics, extraction and use | Concentrated in Asia with some (large) European and US companies | Importance of standards from targeted markets. Large companies dominate production. | Modular, with central role for alginate producers |
| Cosmetics | Growing number of publications Focus on characteristics of seaweed | Most companies in Europe and USA, Asian market is booming | Dominance of few large companies. Many small SMEs. Regulation mainly on ingredients and information provision to consumers. | Modular |

The European sector need to develop ways of how to set itself apart from other producing regions and add value through novel applications. For this to happen the sector should explore and implement an upgrading strategy based on high quality production with strict standards for sustainability and safety. A joint effort to develop safe and sustainable products that meet the demands of regulators, lead-firms and consumers is then needed.

The analysis shows that the five value chains covered here are also quite different seaweed value chains, which will demand a differentiation of the strategy. The main differences lie in the role of the lead-firm, in some value-chain they are very influential, in others not, and the prevailing type of value chain governance. Commonalities include the fact that seaweed production and/or harvesting is done by separate companies that deliver to processors.

In all value-chains, we see that the number of scientific publications increases (a proxy for innovation) and these publications originate from different regions in the world. As innovation in the use of seaweed takes place across the globe, the focus on high-value applications alone will not set the nascent European seaweed sector apart from established

producing regions such as Asia. The organisation of the value-chain does not favour European producers either: a given processor can buy seaweed from various regions. Rules for import and the corresponding safety and environmental standards will be important.

In the further development of the European seaweed industry, more recent statistics would be useful. Recent market information per region and sector/application is often incomplete or a reliable source is missing. Using older information could be misleading as current business models and technologies are developed the last 5–6 years. With [2] we emphasize that data on the amount of algae biomass produced in Europe are fragmented, incomplete and generally of low quality, which prevents a robust quantitative analysis.

Finally, further analysis of value chains can benefit from an in-depth analysis of patents. Although the topic of patents is addressed in [50], further analysis is needed to understand how intellectual property rights influence the further development of the European seaweed sector.

**Author Contributions:** Conceptualization, T.S., E.G. and S.W.K.v.d.B.; methodology, T.S. and S.W.K.v.d.B.; formal analysis, E.G.; investigation, T.S., E.G., S.W.K.v.d.B.; writing—original draft preparation, T.S. and E.G.; writing—review and editing, S.W.K.v.d.B.; funding acquisition, S.W.K.v.d.B. All authors have read and agreed to the published version of the manuscript.

**Funding:** This project has received funding from the European Union's Horizon 2020 research and innovation programme under grant agreement No. 727892 (GENIALG). This output reflects only the author's view and the European Union cannot be held responsible for any use that may be made of the information contained therein.

**Institutional Review Board Statement:** This study did not involve human participants.

**Informed Consent Statement:** This study did not involve human participants.

**Data Availability Statement:** Not Applicable.

**Conflicts of Interest:** The authors declare no conflict of interest.

## Appendix A. Number of Search Results Supplementary

**Table A1.** Number of search results for each value chain.

|  | Seaweed OR Macro-Algae OR Macroalgae OR Brown Macro-Algae OR Green Macro-Algae OR Brown Macroalgae OR Green Macroalgae OR Saccharina Latissima OR Ulva |
| --- | --- |
| Pharmaceuticals | 732 |
| Pharmaceuticals and Fucoxanthin | 26 |
| Bioplastic OR bio-plastic | 16 |
| Bioplastic OR bio-plastic OR acrylic acid | 49 |
| Biostimulant OR bio-stimulant | 286 |
| Alginate OR ulvan | 1098 |
| Cosmetics | 406 |
| Cosmetics AND antioxidant OR anti-oxidant | 136 |

## Appendix B. Search Terms Used in Scholar

(i) TITLE-ABS-KEY (pharmaceuticals AND fucoxanthin AND seaweed OR macroalgae OR macro-algae OR "brown macroalgae" OR "brown macro-algae" OR "green macroalgae" OR "green macro-algae" OR "saccharina latissima" OR ulva) AND PUBYEAR > 1989 AND PUBYEAR < 2021. And TITLE-ABS-KEY (pharmaceuticals AND seaweed OR macroalgae OR macro-algae OR "brown macroalgae" OR "brown macro-algae" OR "green macroalgae" OR "green macro-algae" OR "saccharina latissimi" OR ulva) AND PUBYEAR > 1989 AND PUBYEAR < 2021.

(ii) TITLE-ABS-KEY (pharmaceuticals AND fucoxanthin AND seaweed OR macroalgae OR macro-algae OR "brown macroalgae" OR "brown macro-algae" OR "green macroalgae" OR "green macro-algae" OR "saccharina latissimi" OR ulva) AND PUBYEAR > 1989 AND PUBYEAR < 2021.

(iii) TITLE-ABS-KEY (bioplastic OR bio-plastic OR "acrylic acid" AND seaweed OR macroalgae OR macro-algae OR "brown macroalgae" OR "brown macro-algae" OR "green macroalgae" OR "green macro-algae" OR "saccharina latissimi" OR ulva) AND PUBYEAR > 1989 AND PUBYEAR < 2021. And TITLE-ABS-KEY (bioplastic OR bio-plastic AND seaweed OR macroalgae OR macro-algae OR "brown macroalgae" OR OR "brown macro-algae" OR "green macroalgae" OR "green macro-algae" OR "saccharina latissimi" OR ulva) AND PUBYEAR > 1989 AND PUBYEAR < 2021.

(iv) TITLE-ABS-KEY (bioplastic OR bio-plastic OR "acrylic acid" AND seaweed OR macroalgae OR macro-algae OR "brown macroalgae" OR "brown macro-algae" OR "green macroalgae" OR "green macro-algae" OR "saccharina latissima" OR ulva) AND PUBYEAR > 1989 AND PUBYEAR < 2021.

(v) TITLE-ABS-KEY (biostimulant OR bio-stimulant AND seaweed OR macroalgae OR macro-algae OR "brown macroalgae" OR "brown macro-algae" OR "green macroalgae" OR "green macro-algae" OR "saccharina latissimi" OR ulva) AND PUBYEAR > 1989 AND PUBYEAR < 2021.

(vi) TITLE-ABS-KEY (biostimulant OR bio-stimulant AND seaweed OR macroalgae OR macro-algae OR "brown macroalgae" OR "brown macro-algae" OR "green macroalgae" OR "green macro-algae" OR "saccharina latissimi" OR ulva) AND PUBYEAR > 1989 AND PUBYEAR < 2021.

(vii) TITLE-ABS-KEY (alginate OR ulvan AND seaweed OR macroalgae OR macro-algae OR "brown macroalgae" OR "brown macro-algae" OR "green macroalgae" OR "green macro-algae" OR "saccharina latissima" OR ulva) AND PUBYEAR > 1989 AND PUBYEAR < 2021.

(viii) TITLE-ABS-KEY (alginate OR ulvan AND seaweed OR macroalgae OR macro-algae OR "brown macroalgae" OR "brown macro-algae" OR "green macroalgae" OR "green macro-algae" OR "saccharina latissima" OR ulva) AND PUBYEAR > 1989 AND PUBYEAR < 2021.

(ix) TITLE-ABS-KEY (cosmetics AND seaweed OR macroalgae OR macro-algae OR "brown macroalgae" OR "brown macro-algae" OR "green macroalgae" OR "green macro-algae" OR "saccharina latissima" OR ulva) AND PUBYEAR > 1989 AND PUBYEAR < 2021.

(x) TITLE-ABS-KEY (cosmetics AND seaweed OR macroalgae OR macro-algae OR "brown macroalgae" OR "brown macro-algae" OR "green macroalgae" OR "green macro-algae" OR "saccharina latissima" OR ulva) AND PUBYEAR > 1989 AND PUBYEAR < 2021.

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
