# Peer review of "Innovation and Collaboration: Opportunities for the European Seaweed Sector in Global Value Chains"

_jmse, doi:10.3390/jmse9070693_

Round 1

Reviewer 1 Report

In my opinion, the work was constructed in a thoughtful and matter-of-fact manner. The presented issues are interesting from a scientific point of view and constitute a significant contribution to the development of the discipline. The theoretical background has been described very clearly and legibly. The authors conducted a thorough analysis of the literature related to the discussed topic. It is worth emphasizing both the quantity and quality as well as the topicality of the cited literature. As many as 71% of items (32 out of 45) come from 2015 or newer.
I recommend acceptance of the article and its publication.

Author Response

We thank the reviewer for the positive feedback.

Reviewer 2 Report

This is an interesting paper on the prospective future of seaweed production/applications in Europe, based on the framework developed by Gereffi and others.

I must state that I am a phycologist, not an economist, thus, my comments will be on my knowledge on seaweeds biology, cultivation and bioactivities, and their possible use in different sectors of the industry.  

Although I understand that this paper was written by economists, it is integrated in GENIALG project. Thus, in the introduction, a description of the seaweeds Saccharina latissima and of Ulva spp. is required:

“Saccharina latissima (Linnaeus), is a brown seaweed known as sugar kelp or kombu royal in Europe. It is the most cultivated species in the world, and in Europe, for food (check The State of World Fisheries and Aquaculture 2020 for data on cultivation). Ulva spp., known as sea lettuce, is a group of green seaweeds, also presenting high biomass production yield and showing high farming expansion potential in Europe.”

https://doi.org/10.1016/j.seppur.2020.117589

https://doi.org/10.1007/s10811-016-0883-7

These are papers published within GENIALG project.

Actually, the only place where these seaweeds are referred to is in the title, which makes no sense. The chosen seaweeds must be addressed to justify why the specific compounds were chosen (e.g. fucoxanthin, alginate or acrylic acid).

Also, regarding, the search, the authors restricted their search to SCOPUS, when there are other scientific search motors complementary, such as Science Direct and Web of Science. These should have been used to expand the search.

Furthermore, I believe that the search performed was too narrow. The authors missed many papers by restricting the search to keywords like “pharmaceuticals”. Many papers published regarding “bioactive compounds” that can be used in the pharmaceutical industry do not specially state that word. Thus, many more papers are being published every year, in different regions, than those identified by the authors. The same for other

Regarding fucoxanthin, alginate, and acrylic acid, the authors must explain why they chose these compounds and not others.

For brown seaweeds, only Fucoxanthin, and alginate were addressed, and for Ulva spp. only acrylic acid that also occurs in some red seaweeds. Ulvans, for example, were not considered for green seaweeds, nor fucoidan or phloroglucinol in brown seaweeds, just to cite a few. The authors must justify why they choose only those three. Also, only acrylic acid was studied, when many other solutions can be found for bioplastic from seaweeds, such as polysaccharides (see e.g. 10.1007/s11356-020-10010-z and 10.30848/PJB2021-4(40)).

So the authors must explain why these compounds/species were chosen and not others.

Finally, in the discussion, I would expect more detail on the commercial issues, such as the number of companies per region, the economic impact of seaweeds per sector, the market value of the products already on the market. The statements are vague throughout the text and not supported in facts or data.

Therefore, I find the paper interesting but incomplete.

I recommend major changes before assessing whether it is fit for publication.

Specific comments are made below to improve the document:

The type of paper is missing.

Title: “spp” is not in italic and a dot is missing in the end.

Introduction

Line 29 – seaweeds produce other phycocolloids besides alginates (Agar, carragenans, ulvans, …), thus the authors should replace “alginates” by “phycocolloids”. The authors may want to specify those produced by the genera chosen: Saccharina (Brown seaweed – Phaeophyceae) that produce alginates and Ulva (Green seaweed – Chlorophyta) which produces ulvans.

Line 50 – again, the authors must specify in the objectives the species/genera they are addressing, as they did in the title. Because they are not addressing all seaweeds (as can be seen in Ap. 2).

Line 83 - Line 2.2 – Operationalisation

Being in May 2021, I can’t understand why the year 2020 is not included.

Figure 2 – improve the quality. The same for other figures.

Line 141 – caption: what does the “i” stand for? The same for other figures ii, iii, iv, …)

Line 154 – the data on seaweed aquaculture are on the FAO’s annual report: The State of World Fisheries and Aquaculture 2020

Line 171 – exactly the same reasoning is argued by the authors Pinteus et al (https://doi.org/10.1016/j.algal.2018.06.018) regarding Asparagopsis armata, but I really don’t understand this reference.
Should the wild harvest of invaders be considered as an opportunity?

Line 188 - The paper Current Status of the Algae Production Industry in Europe: An Emerging Sector of the Blue Bioeconomy 10.3389/fmars.2020.626389 may give the authors an interesting insight on the seaweed sector.

Line 313. When discussing phycocolloids, the authors must explain why they chose only alginate. Europe has a well-established industry of agar, another phycocolloid extracted from red seaweeds (agarophytes), some of them widely cultivated. Thus, it is hard to understand why the authors chose specifically alginate and not the others.

Line 322. The authors may also discuss if these species are harvested or cultivated.

Line 326. I found 41,271 papers on alginates (the word stated in the caption of figure 8) in SCOPUS. The search the authors performed is too narrow I believe.

Try to explain the search performed and why.

Line 383. Ulvan Polysaccharides from green seaweeds are also important compounds for cosmetic purposes. Check e.g. https://doi.org/10.3390/biom10070991

Line 416. Powerful statements like “However, the Asian cosmetics market is booming.” have no justification or citation.

Line 444 to 449 – give further detail on sectors/regulations already implemented and on the making. This analysis is too shallow to be useful.

Line 460 – table 2 – on what do the authors support the claim that seaweeds play a marginal role in bioplastics?

Line 478. Will these strict regulations, thus, foster European seaweed production in aquaculture?

Author Response

for a reply to reviewers, see attached document

Reviewer 3 Report

The manuscript entitled: "Innovation and collaboration: the opportunities for European Saccharina latissima and Ulva spp value chains" presents an interesting approach by presenting an analysis of 5 value chains, aspect that is not well covered in the literature. In general, I see this paper published in JMSE soon. My major criticism is that the authors base its analysis on scientific publications but maybe the existing patents could offer a mor realistic perspective from a value chain point of view (only in the Conclusion section the authors mention that publications were used as a proxy for innovation). As this analysis is not done by the authors, they need to discuss this limitation adding one or two sentences. In addition, and more relevant there are several points within the presentation of the paper that requires attention which are the following:

1) Introduction: Due to the scope and reader expertise of JMSE, I have the perception that the paper needs some general seaweed productive trends description. I suggest that some of this can be found in Buschmann et al. 2017 Eur. J. Phycol. and Naylor et al 2021 Nature, Chopin & Tacon 2020 Rev. Fish. Sci. and Aquacul. This background information seems necessary from my point of view to a correct understanding of the ideas developed by the authors that makes sense for a broad audience.

2) Introduction:  I agree with that food chain systems analysis can provide valuable information for the development of this industry in Europe. However, as this is the specific topic, I think it is relevant to indicate what are the main chain roads of seaweeds are at the present. I do not expect a detailed analysis just a brief overview of the most relevant food chain. 

3) Materials and Methods (section 2.2 Operationalisation): Please add the searching words and the date of the search for each selected value chain (example: Data collection was carried out by searching in in Google scholar (line 672) or Scopus (line with following keyword: ......; search date: ....). The authors can also cite the Appendix B with the specific search terms. In addition, I would fuse the next section 2.3 Data collection as this information can be added as provided in the example. By reviewing appendix B it looks that the search terms are not specific for Saccharina or Ulva? This point makes the interpretation of the data not to be related to both species and Europe. This requires to be better presented in M&M.

4) Materials and Methods: the authors must also indicate how they manage review and other kind of papers. Is paper based only on original research paper? Another question, the search was for seaweed or macroalgae plus pharmaceutic and the other keywords, or restricted for Saccharina and Ulva?  Please clarify.

5) Results: I could not see in this section the emphasis on Ulva and Saccharina as the title suggest. For example, in section 3.1. Pharmaceutical I do not see the emphasis on Saccharina at all. It looks like an analysis for brown algae in general and the authors never indicate anything for the title indicated brown algae. Even in paragraph presented on lines 163-170 they discuss the potential use of Sargassum, but not Saccharina.  Should the authors change the title or focus on the subject in detail.

6) Results (section 3.2 Bioplastics, Line 199): The authors indicate "Seaweed is a candidate ... ", but the term seaweed are represented by phylogenetically very different type of organisms: all are useful for bioplastic production? Again, this section seems general for any type of seaweed and I cannot fallow the restriction of Saccharina and Ulva in regard to this value chain.

7) Results (section 2.3 Bioestimulants): I have the same comments as for the previous value chain analysis.

8) Discussion and Conclusion section: I do not see anything that focus on Saccharina and Ulva presented in the title.

9) I see several statements that have no backup numbers or references. Some examples, but please check the whole paper to be fully consistent with your statements and conclusions: 1) line 419-420), which companies, where is the data to reach this conclusion? 2) lines 425-426, the authors indicate "The market in the United States and Japan is however dominated by few large companies". Where is this information coming? 3) line 369-370, where is the data for alginate companies? 4) check the complete text carefully I see several additional issues like the ones in the examples

10) Figure captions: the authors indicate as a superscript at the end of the fire legend were the search term can be found. The authors need to be more explicit and add at the end of the figures: “Search term in Appendix B: i.”

11)  The logic flow of the Discussion be reviewed.  First, I would avoid the subsection and provide a unified discussion section. As the authors start discussion the comparison of the food chain it seems that the authors must start with the most general issue presented in the 3rd paragraph (lines 442-449. After this the authors can discuss how new studies are providing new insight, governance for the industry. In general please provide a better structured Discussion section and also include some recent relevan published review papers (e.g. Wells et al. 2017 JAPH, Shannon & Abu-Ghannam 2019, Phycology Holdt and Kraan 2011 JAPH, between several others) on some of these subjects to provide a more robust context to your findings. The paper must be explicit for each reader what is the novelty of this results and think that this journal is reviewed not only by seaweed experts.

12) Tables and Figures citation in the text must be checked. The result summary is presented on Table 2, but this table is not cited (search in the pdf file was carried out) in any point of the text. It seems required to be cited in the Discussion.

Note: The quality of the figures was not good. I do not know if the explanation is related to the journal platform or the authors needs to improve the resolution of their images. Please verify the resolution, I could hardly read the figures.

Author Response

(The authors gave the same response as above.)

Round 2

Reviewer 2 Report

I  acknowledge the extensive changes made by the authors to improve the paper. Most of the suggestions were accepted and the paper was altered accordingly. Extensive explanations regarding my questions were delivered and were clear.

The suggestions that were not incorporated in the text are well explained and I accept them, most of them being due to the project's GENIALG choices.

The methods used were better explained, and thus are clearer.  Also the discussion has been improved.

Thus, I recommend the publishing of the paper with minor changes. 

Check the taxonomic rules: scientific names in italic, genus epithet in capital letter and species epithet in lower case letter: Saccharina japonica and Ulva spp. In lines 48 and 49: Spp. Shouldn’t be in italic, only the scientific names must.

Line 107: I believe that ulvans should be added, for the authors included these compounds in the search

Line 128: latissima is not written in capital letter

Author Response

L.s.

thanks for the reply. We are happy to see our edits are appreciated by the reviewers. The minor changes mentioned below  have been made in the 2nd revision of the document.

Check the taxonomic rules: scientific names in italic, genus epithet in capital letter and species epithet in lower case letter: Saccharina japonica and Ulva spp. In lines 48 and 49: Spp. Shouldn’t be in italic, only the scientific names must.

--> done. This also led us to change the citation (first page, left hand site)

Line 107: I believe that ulvans should be added, for the authors included these compounds in the search

--> correct and added

Line 128: latissima is not written in capital letter

--.> changed